# Accumulation of Alpha-Synuclein and Increase in the Inflammatory Response in the *substantia nigra*, Jejunum, and Colon in a Model of O_3_ Pollution in Rats

**DOI:** 10.3390/ijms25105526

**Published:** 2024-05-18

**Authors:** Marlen Valdés-Fuentes, Erika Rodríguez-Martínez, Selva Rivas-Arancibia

**Affiliations:** Departamento de Fisiología, Facultad de Medicina, Universidad Nacional Autónoma de México, Mexico City 04510, Mexico; marlen_valdes@ciencias.unam.mx (M.V.-F.); e.rodriguez@comunidad.unam.mx (E.R.-M.)

**Keywords:** ozone pollution, neurodegeneration, alpha-synuclein, inflammation, NFκB, Th17, *substantia nigra*, gut

## Abstract

This work aimed to study the effect of repeated exposure to low doses of ozone on alpha-synuclein and the inflammatory response in the *substantia nigra*, jejunum, and colon. Seventy-two male Wistar rats were divided into six groups. Each group received one of the following treatments: The control group was exposed to air. The ozone groups were exposed for 7, 15, 30, 60, and 90 days for 0.25 ppm for four hours daily. Afterward, they were anesthetized, and their tissues were extracted and processed using Western blotting, immunohistochemistry, and qPCR. The results indicated a significant increase in alpha-synuclein in the *substantia nigra* and jejunum from 7 to 60 days of exposure and an increase in NFκB from 7 to 90 days in the *substantia nigra*, while in the jejunum, a significant increase was observed at 7 and 15 days and a decrease at 60 and 90 days for the colon. Interleukin IL-17 showed an increase at 90 days in the *substantia nigra* in the jejunum and increases at 30 days and in the colon at 15 and 90 days. Exposure to ozone increases the presence of alpha-synuclein and induces the loss of regulation of the inflammatory response, which contributes significantly to degenerative processes.

## 1. Introduction

Environmental pollution, particularly ozone (O_3_), can cause oxidative stress in the body. Ozone is a gas that has highly oxidizing properties. While it protects living beings from ultraviolet radiation in the stratosphere, it can cause chronic oxidative stress when repeatedly inhaled at the troposphere’s level. This can lead to an increase in the number of reactive oxygen species (ROS) in the body. Prolonged exposure to ozone is linked to the development of respiratory [1], gastrointestinal [2], and nervous system disorders [3]. Under normal physiological conditions, antioxidant systems effectively neutralize an increase in ROS [4]. However, these antioxidant systems cannot counteract repeated exposure to low doses of O_3_, equivalent to a day of high pollution. Due to the inhalation of this gas on days of high environmental pollution, ROS are formed mainly in the lungs and enter the circulation; another entry route is the olfactory epithelium through chemoreceptors that communicate directly with the olfactory bulb and, from there, to other brain areas [5,6,7], entering directly into the brain. When ROS enter the body and spread through the bloodstream, they disrupt the balance of oxidants and antioxidants. This disruption stimulates the production of cytokines and proteins that participate in signaling cascades, promoting an inflammatory process [8,9,10]. ROS also affect genetic and epigenetic mechanisms, such as the transcription of certain genes that encourage the immune response [11,12,13]. Additionally, degenerative diseases, such as Alzheimer’s, Parkinson’s, and Huntington’s disease, involve the misfolding and aggregation of proteins like beta-amyloid, phosphorylated tau, and alpha-synuclein (α-Syn).

α-Syn is a protein found in the cytoplasm in an unfolded form. It can adopt more stable structures by folding [14]. The protein is involved in microglia activation and is linked to dopamine metabolism [15,16,17]. Although it is associated with vesicle transport and presynaptic functions, its exact role in the cell remains unclear. However, when the protein misfolds, it induces the homo-oligomerization of α-Syn, which results in pathological aggregation [18]. Moreover, the aggregation of α-Syn directly triggers inflammatory pathways, leading to an increased secretion of the cytokine, IL-1β, by activating inflammasome pathways, such as the pyrin domain of the NLR family (NLRP3) [19]. Furthermore, it has been proposed that the induced and prolonged inflammation causes neuronal degeneration [20], making neuronal death imminent. On the other hand, a synucleinopathy has been reported; works published on rats show that when α-Syn fibrils are seeded in the intestine [21] or through an intramuscular application [22], the neurodegenerative pathology progresses in the brain. In 2005, Luk, Song, and O’Brien reported that seeding α-Syn recruits other endogenous α-Syn proteins, which are converted into insoluble pathological proteins through hyperphosphorylation and ubiquitination [23]. It has been widely documented that there is a close relationship between the gastrointestinal tract and the brain, which is referred to as the gut–brain axis. This communication occurs through neural and hormonal signals that are generated by the gastrointestinal epithelium and the central nervous system [24,25]. Communication between the gut and the brain involves immune cells from the mucosa within the intestinal epithelium and other cells in the intestinal environment, which are part of the enteric nervous system [26,27]. In this regard, it has been proposed that other factors in the intestinal microenvironment, such as the microbiota, play a crucial role in comprehending the pathogenesis of certain diseases, especially neurodegenerative diseases [28]. Intestinal dysbiosis occurs when there is an imbalance in the quantity or types of microbial communities, leading to increased intestinal permeability and inflammation [29].

On the other hand, the loss of intestinal permeability regulation is linked to neurodegenerative diseases through cellular and molecular mechanisms. The study of molecules such as cytokines, hormones, and proteins that act as mediators between the immune system, the brain, and the intestine is crucial to understanding degenerative pathologies. The misfolding of the α-Syn protein can lead to synucleopathies, which are a group of conditions that include multiple systemic atrophies, Parkinson’s disease, and dementia with Lewy bodies. These diseases are identified by the presence of inclusion bodies in neurons and glia, which are formed by aggregates of insoluble synuclein proteins [30].

During the inflammatory process, immune system cells are recruited to produce a specific response. One of the mechanisms involved in regulating the magnitude and duration of the inflammatory response is the Th17 response [31]. Th17 cells have a significant role in various inflammatory diseases by secreting the cytokine IL-17 in response against extracellular bacteria, fungi, and viruses. The plasticity of Th17 cells is influenced by various factors, such as cytokines and microbial and metabolic products [32]. Additionally, the immune response is regulated by activating specific transcription factors. The Th17 response triggers the activation of the nuclear factor *kappa*B (NFκB) transcription factor [33,34], which can induce inflammation and oxidative stress by transcribing certain genes [35]. It has been recently reported that α-Syn degradation occurs through autophagy. This process is facilitated by the presence of spike-like receptor 4 (TLR4), which interacts with α-Syn and enables the transcription of specific genes through NFκB [36]. A study conducted in 2023 by Jackson et al. showed that NFκB signaling upregulates inflammation-associated genes in an acute ozone model [37].

Therefore, our interest lies in studying how repeated exposure to low doses of ozone causes alterations in α-Syn and changes in NFκB and IL-17 both in the *substantia nigra* and the jejunum and colon and how these alterations participate in degenerative diseases.

## 2. Results

### 2.1. Oxidative Stress Generates an Increase in a-Syn in the substantia nigra, Jejunum, and Colon

Immunohistochemistry for α-Syn in the *substantia nigra* showed an increase in the protein (Figure 1A). The protein labeling is observed to be more abundant within the cells at 15 (Figure 1c), 30 (Figure 1d), and 90 days of exposure to O_3_ (Figure 1f) concerning the group control. The expression of the *Snca* gene showed a significant increase (* *p* ≤ 0.05) at 15 (N = 4; 95% CI = 1.233–8.196), 60 (N = 4; 95% CI = 2.114–4.349), and 90 days of O_3_ exposure (N = 4; 95% CI = 3.159–11.08) (Figure 1B), while the relative abundance for the α-Syn protein showed a significant increase (* *p* ≤ 0.05) at 7 days (N = 6; 95% CI = 0.189–598), 15 (N = 6; 95% CI = 2.45–9.97), 30 (N = 6; 95% CI = 1.67–8.23), and 60 days of exposure (N = 6; 95% CI = 2.40–7.84) (Figure 1C).

Immunohistochemistry for α-Syn in the jejunum increased the protein (Figure 2A). The protein labeling is observed to be more abundant within the cells at 15 (Figure 2c), 60 (Figure 2e), and 90 days of exposure to O_3_ (Figure 2f) concerning the group control. The expression of the *Snca* gene showed a significant increase (* *p* ≤ 0.05) at 15 (N = 4; 95% CI = 3.93–8.58), 60 (N = 4; 95% CI = 2.193–5.57), and 90 days of O_3_ exposure (N = 4; 95% CI = 1.96–2.78) (Figure 2B), while the relative abundance for the α-Syn protein showed a significant increase (* *p* ≤ 0.05) at 7 days (N = 6; 95% CI = 0.37–1.61), 15 (N = 6; 95% CI = 0.02–0.65), 30 (N = 6; 95% CI = 0.007–0.48), and 60 days of exposure (N = 6; 95% CI = 0.23–1.27) (Figure 2C).

Immunohistochemistry for α-Syn in the colon showed an increase in the protein (Figure 3A). The protein labeling is more abundant in the cells at 30 (Figure 3d) and 60 days of O_3_ exposure (Figure 3e) with respect to the control group. The expression of the *Snca* gene showed a significant increase (* *p* ≤ 0.05) at 15 (N = 4; 95% CI = 0.71–7.05), 30 (N = 4; 95% CI = 11.67–23.86), 60 (N = 4; 95% CI = 1.51–6.29), and 90 days of O_3_ exposure (N = 4; 95% CI = 0.63–1.35) (Figure 3B), while the relative abundance for the α-Syn protein did not show significant changes during treatment (Figure 3C).

### 2.2. Rela Expression under Exposure to Low Doses of O_3_

The expression of the gene *Rela* as well as the relative abundance of NFκB are shown in Figure 4. The expression of the gene in the *substantia nigra* showed a significant increase at 7 (N = 4; 95% CI = 2.29–7.67), 15 (N = 4; 95% CI = 5.50–11.51), 30 (N = 4; 95% CI = 7.71–9.53), and 60 days (N = 4; 95% CI = 7.29–10.35) concerning the control group (Figure 4A), while the relative abundance had a significant increase at 7 (N = 6; 95% CI = 1.36–2.41), 15 (N = 6; 95% CI = 0.55–2.30), and 30 days (N = 6; 95% CI = 0.52–1.39) compared to the control group (Figure 4B). The expression of the gene in the jejunum showed a significant increase at 7 (N = 4; 95% CI = 3.34–5.27), 30 (N = 4; 95% CI = 4.13–5.85), 60 (N = 4; 95% CI = 1.42–3.38), and 90 days of exposure (N = 4; 95% CI = 0.90–4.93) (Figure 4C). The NFκB protein in the jejunum increased significantly at 7 (N = 6; 95% CI = 0.70–0.99) and 15 days (N = 6; 95% CI = 0.08–0.78) and a subsequent decrease at 90 days (N = 6; 95% CI = 0.04–0.19) compared to the control group (Figure 4D). Finally, the colon gene expression increased significantly at all exposure times, 7 (N = 4; 95% CI = 6.84–10.36), 15 (N = 4; 95% CI = 5.57–18.05), 30 (N = 4; 95% CI = 13.05–33.01), 60 (N = 4; 95% CI = 9.58–12.71), and 90 (N = 4; 95% CI = 6.11–8.84) (Figure 4E), while the abundance of the protein presented a significant decrease at 60 (N = 6; 95% CI = 0.73–1.64) and 90 days (N = 6; 95% CI = 0.93–2.65) compared to the control group (Figure 4F) (Mann–Whitney U test *p* ≤ 0.05).

### 2.3. Effect of O_3_ Exposure to Low Doses on Interleukin IL-17

The expression of the *IL-17* gene in *Substantia nigra* showed an increase at 7 (N = 4; 95% CI = 1.41–2.50), 15 (N = 4; 95% CI = 1.27–2.98), 60 (N = 4; 95% CI = 2.07–4.78), and 90 days (N = 4; 95% CI = 7.12–9.42) of exposure to low doses of O_3_ (Figure 5A), while the relative abundance of the protein showed an increase significant after 90 days of exposure (N = 6; 95% CI = 0.72–2.80) (Figure 5B). On the other hand, in the jejunum, the gene expression presented a significant increase at 7 (N = 4; 95% CI = 1.17–6.62) and 30 days (N = 4; 95% CI = 4.52–5.97) (Figure 5C), and the relative abundance increased at 30 days after exposure (N = 6; 95% CI = 0.90–2.06) (Figure 5D). In the colon, there was an increase in gene expression at 7 (N = 4; 95% CI = 1.39–3.52), 15 (N = 4; 95% CI = 1.19–4.86), 60 (N = 4; 95% CI = 2.93–5.47), and 90 days (N = 4; 95% CI = 1.33–1.57) (Figure 5E), while the relative abundance of IL-17 in the colon showed an increase at 15 (N = 6; 95% CI = 0.109–0.541) and 90 days of O_3_ exposure (N = 6; 95% CI = 0.1095–0.463) (Figure 5F) (Mann–Whitney U test *p* ≤ 0.05).

## 3. Discussion

Environmental ozone pollution generates ROS, which causes an oxidative stress state and changes in gene expression after activating transcription factors, such as the NFκB factor. This is accompanied by the misfolding of the α-Syn protein. The association of α-Syn with the chronic inflammatory response has been widely described, producing a process of progressive neurodegeneration in the central nervous system [3,38]. The accumulation of α-Syn, mainly in the *substantia nigra*, causes cell death and generates local damage that results in the loss of neurons, as occurs in Parkinson’s disease [39]. Our results show an increase in the α-Syn protein at the level of the *substantia nigra*, as can be observed in the immunohistochemistry for α-Syn (Figure 1A), and also at the level of gene expression (Figure 1B) as well as in the relative abundance of the protein, studied by the Western blot technique (Figure 1C). Some works indicate that the accumulation of α-Syn results from recruiting and converting the unfolded α-Syn protein into pathological fibrils in neurons. These α-Syn fibrils enter healthy neurons through receptor-mediated endocytosis and generate propagation [30], generating subsequent neuronal death [40]. According to our findings, there is a close connection between degenerative diseases and the intestine. Our study reveals the presence of the α-Syn protein in the jejunum (Figure 2) and the colon (Figure 3). The jejunum has the highest presence of this protein. The results indicate that there is an evident accumulation in the jejunum and colon as the ozone exposure time increases (Figure 2A and Figure 3A). Additionally, there is an increase in the expression of the *Snca* gene (Figure 2B and Figure 3A) and the relative abundance of the α-Syn protein (Figure 2C). A 2019 study by Ohlsson and Englund showed ganglionar degeneration in the submucosal and/or myenteric plexus, mainly in the jejunum and colon, in the presence of α-Syn protein deposits [41]. Other studies have shown that the accumulation of α-Syn fibrils leads to the loss of gastrointestinal function, particularly intestinal permeability [42]. This loss contributes to the pathogenesis of certain neurodegenerative diseases, such as Parkinson’s disease [28,43]. Our results show a more significant positive immunoreactivity to the α-Syn protein in the jejunum and colon as the days of ozone exposure pass (Figure 2A and Figure 3A). Furthermore, it has been shown that the accumulation of α-Syn in an A53T (TgM83) transgenic mouse model persistently increased both in the enteric nervous system and the brain, suggesting that the accumulation of α-Syn in both systems arises in parallel [44]. In this sense, in our results, the gradual increase in the α-Syn protein both in *substantia nigra* (Figure 1A–C) and in the intestine (Figure 2A–C and Figure 3A–C) allows us to conclude that there is a parallel increase in both tissues.

There is a hypothesis that suggests that Parkinson’s disease may start in the enteric nervous system and then spread to the central nervous system via the *vagus* nerve [45]. On the other hand, exposure to ozone can generate oxidative stress which leads to an increase in the α-Syn protein. This, in turn, activates transcription factors like the NFκB protein, which play a crucial role in regulating gene transcription in organisms. This regulation, whether positive or negative, enables the expression of proteins that carry out functions such as inflammation, immunity, cell differentiation and growth, and apoptosis [46]. NFκB is a transcription factor that resides in the cytoplasm and gets activated and transported to the nucleus upon receiving a specific stimulus [47]. In our laboratory, we have shown that exposure to ozone causes the translocation of NFκB to the nucleus (article in press), so it can be inferred that oxidative stress participates in the transcription of genes whose function is the maintenance of homeostasis and/or cell death [48], causing the alteration of said responses and altering the regulation of the inflammatory response. In a chronic oxidative environment, the inflammatory response is not self-limiting, leading to the persistence of proinflammatory cytokines, particularly the increase in the Th17 response, which is activated by the effect of ozone [3,8], as well as in response to bacteria and fungi [49], and remains active during the development of autoimmune [50] and neurodegenerative diseases through the secretion of IL17 [51], with a notable decrease in IL-10 [8]. Our results clearly show an increase in interleukin IL17 in *substantia nigra* (Figure 5A,B); in addition, in our work group, the increase in IL-17 in the hippocampus has already been reported using the same study model [8]. However, exposure to environmental ozone pollution in the gastrointestinal tract has not reversed its increase. The relationship between the gastrointestinal tract and the nervous system is a topic that has generated a lot of impact in the scientific community. In patients with Parkinson’s, there is an alteration of the blood–brain barrier (BBB), allowing immune cells to reach the brain. Lan and collaborators in 2022 demonstrated, in an in vitro model, that the integrity of the BBB is altered after treatment with oligomeric α-Syn [52]; therefore, oxidative stress in this condition generates chronic inflammation and consequent neuronal death. Another study using a truncal vagotomy prevented the spread of α-Syn to the brain [53], indicating the close relationship between the gastrointestinal system and the brain. Our results demonstrate a progressive increase in the levels of interleukin IL17 in the *substantia nigra* (Figure 5A,B), as well as in the jejunum (Figure 5C,D) and colon (Figure 5E,F), in relation to the control group.

Our study’s findings indicate that exposure to low levels of ozone can increase the α-Syn protein and IL-17, contributing to a chronic inflammatory response. The study suggests that environmental pollution caused by ozone leads to the generation of ROS, which triggers oxidative stress in cells. This, in turn, causes the accumulation of the α-Syn protein over time (as shown in Figure 1, Figure 2 and Figure 3). Additionally, the study indicates that exposure to ozone leads to an increase in the transcription factor NFκB and interleukin IL-17 in the intestine (as shown in Figure 4). Both T and B cell antigen receptors and Toll-like receptors (TLR) have the ability to activate NFκB [48]. Our results indicate that even though the intestinal microbiota is altered and causes a loss of intestinal permeability, the intestinal walls still respond to ozone through an unregulated inflammatory response. The interaction between the intestine–brain and the immune system can explain the presence of neurodegenerative diseases [27]. Exposure to environmental pollution, particularly O_3_, leads to the generation of ROS in the body. The increase in ROS causes a redox imbalance, which interacts with the immune system, causing changes in the inflammatory response, which loses its regulation. This results in a vicious cycle that can lead to progressive degeneration in both the intestine and the *substantia nigra*, leading to the establishment of degenerative diseases that progress over time, similar to what can occur in patients with Parkinson’s disease. Therefore, the above can be considered a key element in the search for preventive treatments.

## 4. Materials and Methods

### 4.1. Animals

A total of 72 male Wistar rats, weighing around 250 to 300 g, were used for the experiment. They were given food and water as per their requirement, kept in individual acrylic boxes, and maintained under constant temperature and humidity conditions with a 12:12 light–dark photoperiod. All animal maintenance protocols were in compliance with the Official Standard NOM-062-ZOO-1999, known as “Technical specifications for the production, care, and use of laboratory animals”. The Ethics Committee of the Universidad Nacional Autónoma de México, Facultad de Medicina, approved the experiments. The study involved six groups of animals, each receiving a different treatment. The first group was the control group and was exposed to normal air for 30 days. The other five groups were exposed to 0.25 ppm of O_3_ for 4 h daily for 7, 15, 30, 60, and 90 days, respectively. Two hours after the end of each of the experimental groups, the rats were anesthetized with sodium pentobarbital (50 mg/Kg). Each group was then further divided into two subgroups where one subgroup underwent immunohistochemical techniques, followed by the other subgroup going through molecular and biochemical techniques (qPCR and Western blot).

### 4.2. O_3_ Exposure

The method used for exposing low O_3_ doses was based on a technique described by Rivas-Arancibia et al. in 2010 [32]. They used a transparent acrylic chamber with an air diffuser connected to an O_3_ generator. To ensure the O_3_ levels remained constant and controlled throughout the entire experiment, an O_3_ monitor (PCI O3 and Control Systems, West Caldwell, NJ, USA) was used.

### 4.3. Immunohistochemistry

The animals were deeply anesthetized with sodium pentobarbital (50 mg/Kg) (NOM-033-SAG-ZOO-2014) after the brains and intestines were extracted and fixed in 4% paraformaldehyde. The tissues were then dehydrated and embedded in paraffin blocks, and 5 µm thick sections were made and mounted on slides [54]. The tissue samples were deparaffinized, hydrated, and underwent antigen retrieval using (Biocare Medical Concord, Pacheco, CA, USA). Peroxidase activity was inhibited using 3% H_2_O_2_. Blocking was performed to reduce background activity, and an alpha-synuclein antibody (GTX112799, Irvine, CA, USA) was incubated overnight at 4 °C. Biotinylated secondary antibody was used for treatment of the slides (Universal Link, Biocare Medical, CA, USA). The tissue samples were treated with streptavidin (4plus Detection Component, Streptavidin-HRP, Biocare Medical) and a chromogen 3,3-Diaminobenzidine substrate (DAB Kit, ScyTek, West Logan, UT, USA), followed by staining with hematoxylin. The slides were viewed under an BX41 Olympus microscope and photographed using an Evolution-QImagin Digital Camera Kit (MediaCybernetics, Silver Spring, MA, USA) [54].

### 4.4. qPCR

The *substantia nigra*, jejunum, and colon samples were kept in 1 mL of Trizol^®^ to be homogenized. After adding chloroform, isopropanol, and 75% ethanol, the RNA samples were resuspended in 40 μL of nuclease-free water and stored at −70 °C until processing. The RNA quantity and quality were estimated spectrophotometrically at 260–280 nm. A constant amount of RNA was transcribed to cDNA using the SuperScript TM III^®^ (Invitrogen, CA, USA) reverse transcriptase kit. Amplification was performed in triplicate using oligonucleotide sequences listed in Table 1 with an MIC^®^ real-time PCR detection system. The genes were amplified using the Luna Universal qPCR^®^ master mix in a final volume of 25 μL. This was carried out by following an activation, denaturation, and inhibition protocol as reported in Velázquez-Pérez et al. 2021 [8]. The results obtained were analyzed using the 2^−ΔΔCT^ method, with the cycle threshold being normalized with the constitutive gene Rps 18. This helped to calculate the RNA levels of each of the genes.

### 4.5. Western Blot

The *substantia nigra*, jejunum, and colon samples were homogenized, and protein quantification was performed using the BCA method (Micro BCA Protein Assay Kit^®^). Proteins were then separated by electrophoresis on a 10% SDS acrylamide gel under reducing conditions. Blotting was performed using a PVDF membrane and blocked using 5% skim milk in TBST. Primary antibodies for alpha-synuclein (GTX112799), NFκB (GTX102090), and beta-actin (GTX110564) from the GENETEX^®^ brand, as well as IL17 (IL-17 (G-4): sc-374218, SANTA CRUZ BIOTECHNOLOGY, INC., Dallas, TX, USA), were incubated at 1:1000 and 1:500 concentrations overnight at 4 °C. After three washes with TBST, the membranes were incubated with the appropriate secondary antibodies, mouse anti-rabbit, and goat anti-mouse diluted 1:10,000 (SANTA CRUZ BIOTECHNOLOGY, INC. TX, USA). The bands were developed using 1 mL of Immobilon^®^ Forte Western HRP substrate reagent (Millipore^®^) for 1 min and digitized using the GelCapture^®^ program (v 7.0.5. DNR Bio-Imaging System). Densitometric analysis was performed using Image Studio^®^ software (v 5.2.5. LI-COR Bioscience, Lincoln, NE, USA).

### 4.6. Statistics

The statistical analysis was performed using GraphPad Prims^®^ (v5) software. Variance tests were conducted, and non-parametric tests were used based on the population distribution. The Kruskal–Wallis test was used to compare different time points, and multiple pairwise comparisons were made using the Mann–Whitney U test. A *p*-value of ≤0.05 was considered statistically significant for differences between the groups. All data were represented using a box plot.

## 5. Conclusions

In conclusion, the results of this work demonstrate that repeated exposure to low doses of O_3_ causes an accumulation of alpha-synuclein both in the *subtantia nigra* and in the intestine, which is accompanied by an increase in NFkB and an increase in IL-17. All of the above may indicate that exposure to O_3_ causes an increase in pro-inflammatory molecules in the organs studied, which increase with exposure to said gas.

## Figures and Tables

**Figure 1 ijms-25-05526-f001:**
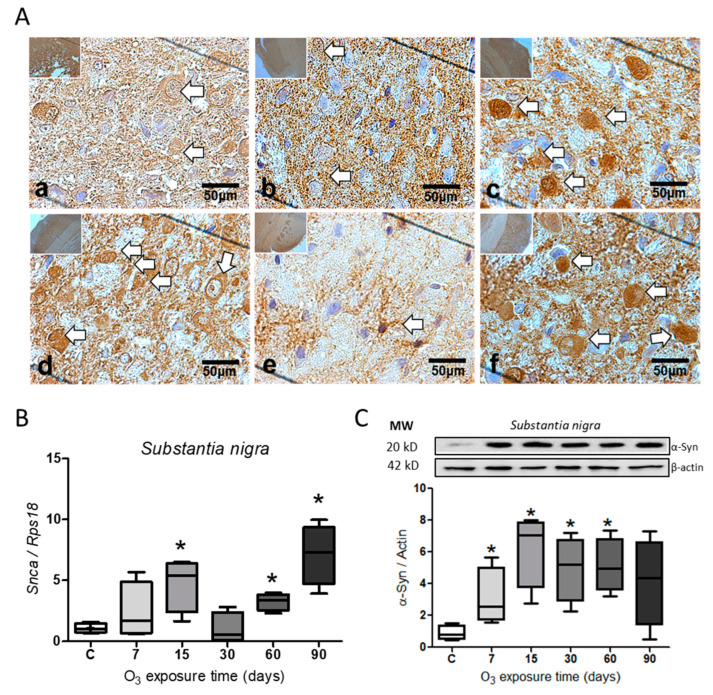
Effect of O_3_ exposure on α-Syn accumulation in *substantia nigra*. (**A**) Micrograph shows (**a**) control, (**b**) 7 days of O_3_, (**c**) 15 days of O_3_ exposure, (**d**) 30 days of O_3_ exposure, (**e**) 60 days of O_3_ exposure, and (**f**) 90 days of O_3_ exposure. Inmunohistochemistry reveals an increase in α-Syn protein after 15, 30, and 90 days of exposure to O_3_; the arrows indicate the immunohistochemistry reaction. (**B**) Gene expression increases significantly at 15, 60, and 90 days of O_3_ exposure. (**C**) The relative abundance shows a significant increase in α-Syn from 7 to 60 days compared to the control group; the images show the representative blot for Western blot band analysis. The graphs show the different treatments to *Snca* gene expression and the relative abundance of α-Syn, represented in arbitrary units. * = *p* ≤ 0.05, the asterisks represent the significant difference compared to the control group.

**Figure 2 ijms-25-05526-f002:**
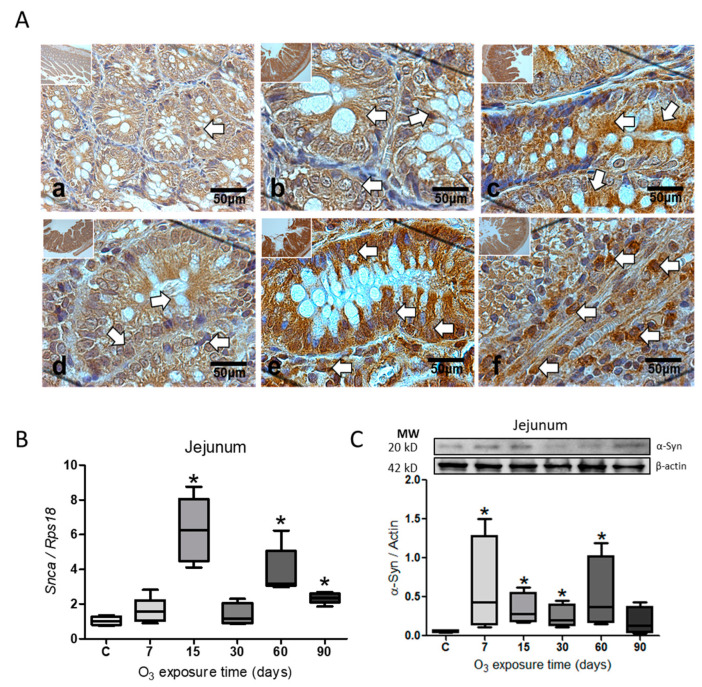
The effect of exposure to O_3_ on the accumulation of α-Syn in the jejunum. (**A**) Micrograph shows in (**a**) control, (**b**) 7 days of O_3_, (**c**) 15 days of O_3_ exposure, (**d**) 30 days of O_3_ exposure, (**e**) 60 days of O_3_ exposure, and (**f**) 90 days of O_3_ exposure. An increase in α-Syn expression is observed at 15, 30, and 90 days of exposure to O_3_. The arrows indicate the immunohistochemistry reaction. (**B**) The gene expression *Snca* analysis shows a significant increase in α-Syn at 15, 30, and 90 days of exposure to O_3_. (**C**). The Western blot band analysis also shows a significant increase in the relative abundance of α-Syn from 7 to 60 days compared to the control group. All data are represented in arbitrary units. The *p*-value for statistical significance is ≤0.05, and the asterisks represent the significant difference compared to the control group.

**Figure 3 ijms-25-05526-f003:**
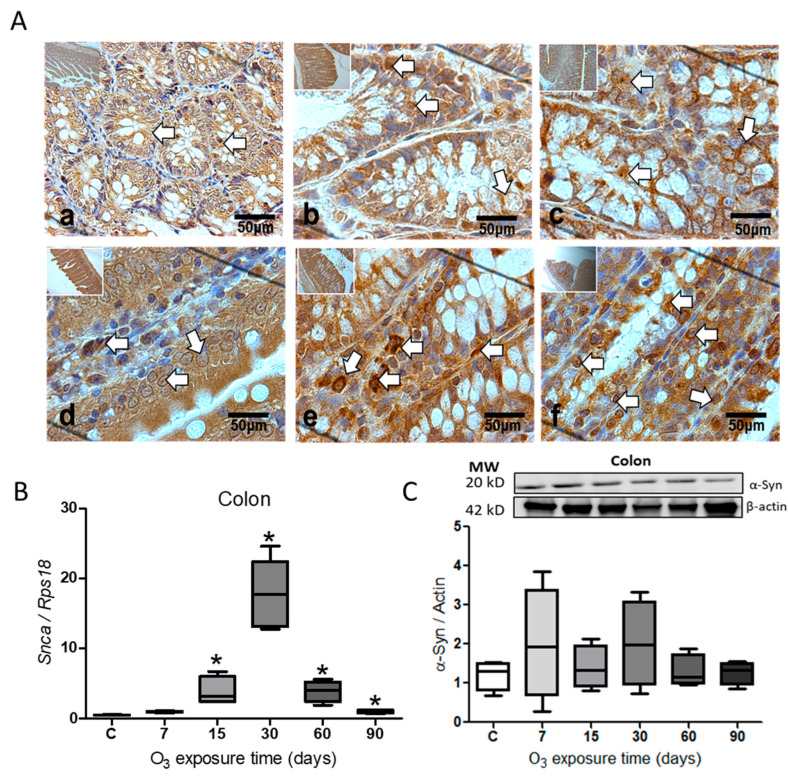
Effect of O_3_ exposure on the accumulation of α-Syn in the colon. (**A**) Micrograph shows in (**a**) control; (**b**) 7 days of O_3_; (**c**) 15 days of O_3_ exposure; (**d**) 30 days of O_3_ exposure; (**e**) 60 days of O_3_ exposure; and (**f**) 90 days of O_3_ exposure. An increase in α-Syn expression is observed after 30 and 60 days of exposure to O_3_; the arrows indicate the immunohistochemistry reaction. (**B**) The gene expression increased significantly from 15 to 90 days of exposure to O_3_. (**C**) The relative abundance did not show significant differences in the treatments; the images show the representative blot for the Western blot band analysis. The graphs show the different treatments to *Snca* gene expression and the relative abundance of α-Syn, represented in arbitrary units. * = *p* ≤ 0.05, the asterisks represent the significant difference compared to the control group.

**Figure 4 ijms-25-05526-f004:**
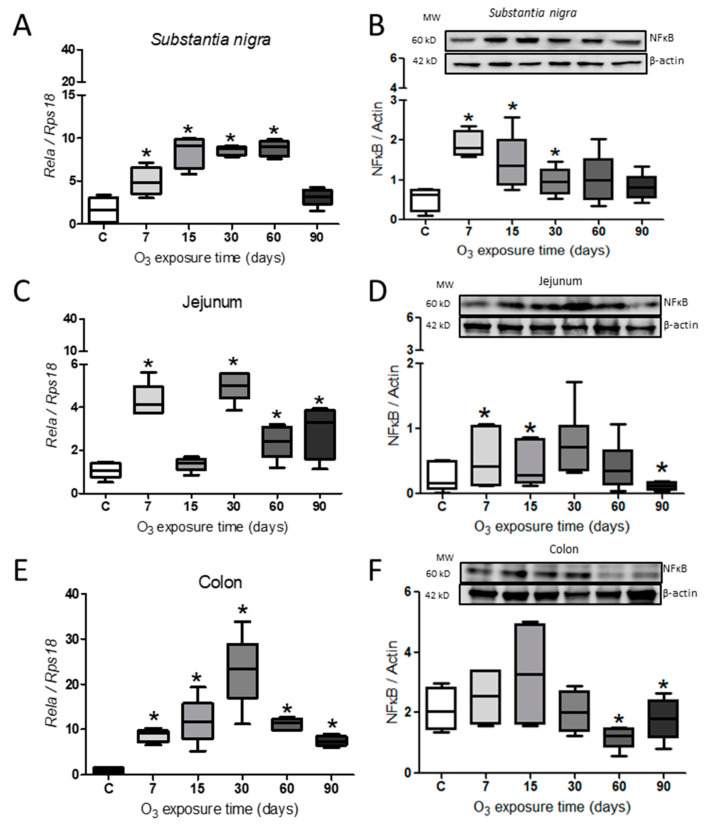
Effect of O_3_ exposure on *Rela* expression. (**A**) Expression of *Rela* in *Substantia nigra*, a progressive increase from 7 days of exposure to 60 days. (**B**) The relative abundance showed a significant increase at 7, 15, and 30 days of exposure to O_3_. (**C**) In the jejunum, there was a significant increase at 7, 30, 60, and 90 days. (**D**) Relative abundance increased at 7 and 15 days and decreased at 90 days. (**E**) In the colon, there was an increase from 7 to 90 days. (**F**) The relative abundance of NFκB in the colon decreased at 60 and 90 days. The images show the representative blot for Western blot band analysis in arbitrary units. Mann–Whitney U test significance of *p* ≤ 0.05 (*) for all treatments, and the asterisks represent the significant difference compared to the control group.

**Figure 5 ijms-25-05526-f005:**
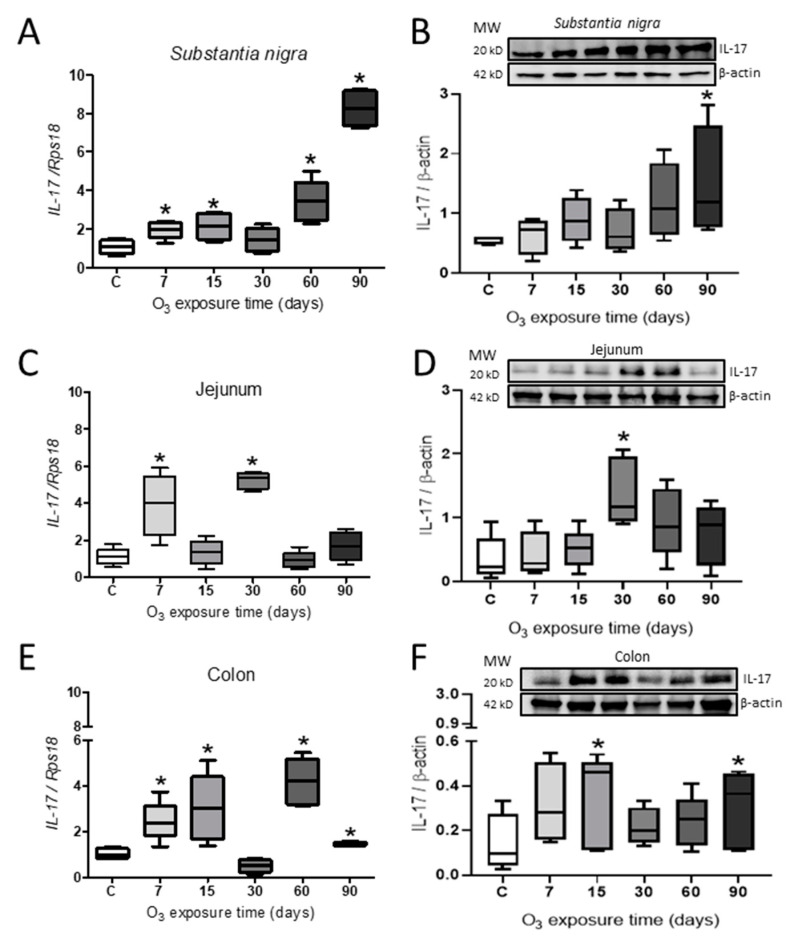
Effect of O_3_ exposure on interleukin IL-17. (**A**) Expression of *IL-17* in *Substantia nigra* increased at 7, 15, 60, and 90 days of exposure. (**B**) Relative abundance showed a significant increase after 90 days of exposure to O_3_. (**C**) There was a significant increase in the jejunum by 7 and 30 days. (**D**) The relative abundance in the jejunum increased after 30 days of exposure. (**E**). In the colon, there was an increase in expression at 7, 15, 60, and 90 days. (**F**) The relative abundance of IL-17 in the colon presents an increase after 15 and 90 days of exposure to O_3_. The images show the representative blot for Western blot band analysis in arbitrary units. Mann–Whitney U test significance of *p* ≤ 0.05 (*) for all treatments, and the asterisks represent the significant difference compared to the control group.

**Table 1 ijms-25-05526-t001:** The oligonucleotide sequences that were used for gene evaluation.

Name	Sequence
*Snca*	Forward: GCCTTTCACCCCTCTTGCAT
	Reverse: TATCTTTGCTCCACACGGCT
*Rela*	Forward: CTT CTG GGC CAT ATG TGG AGA T
	Reverse: TCG CAC TTG TAA CGG AAA CG
*IL17*	Forward: ACT TTC CGG GTG GAG AAG AT
	Reverse: CTT AGG GGC TAG CCT CAG GT
*Rps18*	Forward: TTC AGC ACA TCC TGC GAG TA
	Reverse: TTG GTG AGG TCA ATG TCT GC

## Data Availability

The data presented can be requested by the editors at any time required.

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
