# Peer review of "Accumulation of Alpha-Synuclein and Increase in the Inflammatory Response in the substantia nigra, Jejunum, and Colon in a Model of O3 Pollution in Rats"

_ijms, 2024, doi:10.3390/ijms25105526_

Round 1

Reviewer 1 Report

Comments and Suggestions for Authors

-       It would be more logical to introduce Ozone as O3 while first mentioning in the first sentence.

-       «another entry route is through the olfactory receptors in the nose, entering directly into the brain» this statitment should be supported by a reference. To me this way looks not that obvious for gases. I may be wrong and not updated with the latest data but, regardless the anatomical communication, nose-to-brain route is supposed to have an active component in it as well (rather then to be based on just a chemical diffusion). Additionally, I would replace «olfactory receptors» to «olfactory epithlium» (it’s not only the olfactory but also trigeminal nerve contributed to this transport)

-       Since your study has been performed in rats all the genes should be written in appropriate format which is low case with capital first letter (Snca instead of SNCA). In your paper you used the format specific for human genes.

-       «Immunohistochemistry for α-Syn in the jejunum increased the protein (Figure 2-A).” sounds strange.

-       Somewhere you have used Mann-Whitney criteria for assessment the differences with the control group. The problem is that this method is not eligible when you have more than 2 observed groups. You should go for ANOVA or Kruskal–Wallis test and then for post-hoc analysis implying multiple comparisons corrections.

-       «2.3. Effect of interleukin IL-17 upon exposure to low doses of O3» sounds like IL-17 has shown some effect on O3. It’s the opposite relationships, however (exposure of O3 affected IL-17).

-       « The animals were deeply anesthetized (NOM-033-SAG-ZOO-2014), and the brains and intestines were extracted and fixed in 4% paraformaldehyde with sodium pentobarbital (50 mg/Kg).» Probably, sodium pentobarbital was used for the anesthesia (not for the fixation).

-       Finally, I would recommend to revise the conclusion. You haven’t shown that the observed effects on a-synuclein were related to the inflammation and oxidative stress caused by O3. Maybe those phenomena didn’t have any crosstalk.-       It would be more logical to introduce Ozone as O3 while first mentioning in the first sentence.

-       «another entry route is through the olfactory receptors in the nose, entering directly into the brain» this statitment should be supported by a reference. To me this way looks not that obvious for gases. I may be wrong and not updated with the latest data but, regardless the anatomical communication, nose-to-brain route is supposed to have an active component in it as well (rather then to be based on just a chemical diffusion). Additionally, I would replace «olfactory receptors» to «olfactory epithlium» (it’s not only the olfactory but also trigeminal nerve contributed to this transport)

-       Since your study has been performed in rats all the genes should be written in appropriate format which is low case with capital first letter (Snca instead of SNCA). In your paper you used the format specific for human genes.

-       «Immunohistochemistry for α-Syn in the jejunum increased the protein (Figure 2-A).” sounds strange.

-       Somewhere you have used Mann-Whitney criteria for assessment the differences with the control group. The problem is that this method is not eligible when you have more than 2 observed groups. You should go for ANOVA or Kruskal–Wallis test and then for post-hoc analysis implying multiple comparisons corrections.

-       «2.3. Effect of interleukin IL-17 upon exposure to low doses of O3» sounds like IL-17 has shown some effect on O3. It’s the opposite relationships, however (exposure of O3 affected IL-17).

-       « The animals were deeply anesthetized (NOM-033-SAG-ZOO-2014), and the brains and intestines were extracted and fixed in 4% paraformaldehyde with sodium pentobarbital (50 mg/Kg).» Probably, sodium pentobarbital was used for the anesthesia (not for the fixation).

-       Finally, I would recommend to revise the conclusion. You haven’t shown that the observed effects on a-synuclein were related to the inflammation and oxidative stress caused by O3. Maybe those phenomena didn’t have any crosstalk.

Comments on the Quality of English Language

Some grammar mistakes take place

Author Response

Dear Reviewer 1,

We want to thank you for the corrections made to our work, for correcting our errors, and for allowing us to discuss our ideas.

We have substantially improved this work with your generous time dedicated to our article.

Sincerely,

Selva Rivas-Arancibia

1.- It would be more logical to introduce Ozone as O3 while first mentioning in the first sentence.

Answer.  The initial paragraph of the introduction has been rephrased to make it clearer and easier to understand.

2.- -       «another entry route is through the olfactory receptors in the nose, entering directly into the brain» this statitment should be supported by a reference. To me this way looks not that obvious for gases. I may be wrong and not updated with the latest data but, regardless the anatomical communication, nose-to-brain route is supposed to have an active component in it as well (rather then to be based on just a chemical diffusion). Additionally, I would replace «olfactory receptors» to «olfactory epithlium» (it’s not only the olfactory but also trigeminal nerve contributed to this transport)

Answer. The paragraph of the article was corrected, and references were added to the text.

“ Another entry route is the olfactory epithelium through chemoreceptors that communicate directly with the olfactory bulb and, from there, to other brain areas.”

3.--       Since your study has been performed in rats all the genes should be written in appropriate format which is low case with capital first letter (Snca instead of SNCA). In your paper you used the format specific for human genes.

Answer.- Thank you very much for your correction. The gene names were corrected in the text and the graphs and written correctly

4.-       «Immunohistochemistry for α-Syn in the jejunum increased the protein (Figure 2-A).” sounds strange.

Answer.- In Figure 2, the Western blot graph shows significant differences with the control group for treatments of 7, 15, 30, and 60 days, except for 90 days of ozone exposure.

The representative control group was also changed because of an error in assembling the figure; an image that did not correspond to the treatment was placed.

5.--       Somewhere you have used Mann-Whitney criteria for assessment the differences with the control group. The problem is that this method is not eligible when you have more than 2 observed groups. You should go for ANOVA or Kruskal–Wallis test and then for post-hoc analysis implying multiple comparisons corrections.

Answer. Due to the distribution of the data, we opted for non-parametric tests. Firstly, we used the Kruskal-Wallis test to check if there were any significant differences between all the groups of the different treatments. As we found them to be significant, we further used the Mann-Whitney U test to compare the control group with each of the independent groups of the experiment.

6.- -       «2.3. Effect of interleukin IL-17 upon exposure to low doses of O3» sounds like IL-17 has shown some effect on O3. It’s the opposite relationships, however (exposure of O3 affected IL-17).

Answer.- I appreciate your help. The text has already been corrected.

-       « The animals were deeply anesthetized (NOM-033-SAG-ZOO-2014), and the brains and intestines were extracted and fixed in 4% paraformaldehyde with sodium pentobarbital (50 mg/Kg).» Probably, sodium pentobarbital was used for the anesthesia (not for the fixation).

Answer.- Thank you so much. The text has already been corrected.

“The animals were deeply anesthetized with sodium pentobarbital (50 mg/Kg) (NOM-033-SAG-ZOO-2014), after the brains and intestines were extracted and fixed in 4% paraformaldehyde”

7.- -       Finally, I would recommend to revise the conclusion. You haven’t shown that the observed effects on a-synuclein were related to the inflammation and oxidative stress caused by O3. Maybe those phenomena didn’t have any crosstalk. -       It would be more logical to introduce Ozone as O3 while first mentioning in the first sentence.

Answer.- Exposure to low doses of ozone results in oxidative stress, leading to an imbalance in redox.

An increase in reactive species can affect the immune system. When the redox balance is disrupted, this can trigger a vicious cycle in which the immune system starts producing more reactive species. This can cause inflammation to spiral out of control and lead to a loss of regulation.

The works cited in the references show that alpha-synuclein aggregates activate the immune response.

In our experiment with normal, healthy rats, the only variable was exposure to ozone. This treatment resulted in the accumulation of alpha-synuclein in the substantia nigra, jejunum, and colon. Therefore, I believe the conclusions drawn from this study are likely to be valid.

Reviewer 2 Report

Comments and Suggestions for Authors

The authors exposed rats to ozone under controlled conditions and investigated exposure dependent changes in expression levels of RNA and proteins representing the gut-brain inflammation axis (inflammosome pathway). They report an increase in alpha-synuclein protein levels and dysregulation of the inflammatory response after prolonged (7-90 days)  O3 exposure.

This paper is interesting and appears to be quite well performed in terms of experimental methods.

The selection of proteins for analysis is possibly not the very best for representation of the gut-brain axis as alpha-synuclein is probably not the first one that comes in mind here, but rather caspase 11, NLRP3/AIM, gasdermin D and IL-18 and IL-1ß. However, there are maybe specific reasons to focus on the proteins selected by the authors. The link to Parkinson 's disease is rather weak, as the involvement of the gut-brain axis in this context is still hypothetical. Please explain.

In the introduction, the authors present a paragraph about Th17 cells. While it is very likely known to many readers that Th17 cells are linked to IL-17, the authors should first explain why Th17 cells are of interest in this experimental setup.

Most of the expression levels shown in the paper (except IL-17 in substantia nigra) do not follow a linear exposure time path. Assuming increasing tissue O3 levels and very likely also ROS levels, what are the explanations for reduced levels after prolonged exposure, sometimes even to or below control levels (colon IL-17/SRP18). Please explain.

Some essential information are missing that need to be added to the text of each according figure: Every depiction of statistical data needs a description in the form of (N (number of independent experiments used for calculation), +/-1,5xIQR (or other whisker definition) and significance indicators (as already provided in current version) (like (N= 6, whisker: 1.5xIQR, p ≤ 0.05 (*). Box blot whiskers are not necessarily defined as 1.5xIQR after Tukey.

The English language of the manuscript is quite good. However, writing and language should be improved before publication. Some examples are provided in detail below.

line 13: fullstop after ppm

line 19:.. increases the presence of alpha-synuclein and induces loss of….

line 26: Ozone is not a simple environmental pollutant. Its presence in the troposphere is essential as UV-shield and some O3 is also detected under natural conditions. O3 is however also not the general “main” inducer of ROS. Please rephrase sentence.

line 35: Systemic uptake of relevant levels of O3 by “olfactory receptors”directly to the brain needs to be supported by literature.

line 42: It is clear that alpha-Syn means alpha-synuclein. Please nevertheless explain every abbreviation at first use.

line 45: microglia and dopamine: literature required

line 47: induces homooligomerization

line 51: that the induced and prolonged inflammation causes neuronal degeneration..

line 54: ..the neurodegenerative pathology progresses in the brain.

line 77 “emit” is a strange word here

line 97,116: Immunhistochemistry revealed an increase of alpha-syn protein after..

line 269: In the view that there are many more proteins involved, I would suggest to write “contributing to inflammatory response”

line 329: EE.UU ??

Comments on the Quality of English Language

The English language of the manuscript is quite good. However, writing and language should be improved before publication. Some examples are provided in detail below.

line 13: fullstop after ppm

line 19:.. increases the presence of alpha-synuclein and induces loss of….

line 26: Ozone is not a simple environmental pollutant. Its presence in the troposphere is essential as UV-shield and some O3 is also detected under natural conditions. O3 is however also not the general “main” inducer of ROS. Please rephrase sentence.

line 35: Systemic uptake of relevant levels of O3 by “olfactory receptors” directly to the brain needs to be supported by literature.

line 42: It is clear that alpha-Syn means alpha-synuclein. Please nevertheless explain every abbreviation at first use.

line 45: microglia and dopamine: literature required

line 47: induces homooligomerization

line 51: that the induced and prolonged inflammation causes neuronal degeneration..

line 54: ..the neurodegenerative pathology progresses in the brain.

line 77 “emit” is a strange word here

line 97,116: Immunhistochemistry revealed an increase of alpha-syn protein after..

line 269: In the view that there are many more proteins involved, I would suggest to write “contributing to inflammatory response”

line 329: EE.UU ??

Author Response

Dear reviewer,

We are immensely grateful for the corrections you provided for our work. Your input has been incredibly valuable in helping us to correct our mistakes, clarify our article, and enhance our methodology and discussion. As authors, we would like to express our sincere appreciation for the time and effort you have devoted to improving our manuscript.

Thank you very much.

Sincerely

Dr. Selva Rivas-Arancibia

The authors exposed rats to ozone under controlled conditions and investigated exposure dependent changes in expression levels of RNA and proteins representing the gut-brain inflammation axis (inflammosome pathway). They report an increase in alpha-synuclein protein levels and dysregulation of the inflammatory response after prolonged (7-90 days) O3 exposure.

This paper is interesting and appears to be quite well performed in terms of experimental methods.

1.- The selection of proteins for analysis is possibly not the very best for representation of the gut-brain axis as alpha-synuclein is probably not the first one that comes in mind here, but rather caspase 11, NLRP3/AIM, gasdermin D and IL-18 and IL-1ß. However, there are maybe specific reasons to focus on the proteins selected by the authors. The link to Parkinson 's disease is rather weak, as the involvement of the gut-brain axis in this context is still hypothetical. Please explain.

Answer. Your proposed selection of proteins, including caspase 11, NLRP3/AIM, gasdermin D, IL-18, and IL-1ß, is very interesting. We have previously studied IL-1ß and caspase 3 but have not yet studied related proteins associated with the inflammasome and pyroptosis. However, we plan to begin experiments with these proteins shortly.

Numerous articles establish a connection between the gut and brain, known as the "gut-brain axis" (references 21, 22, 25, 26, 27 and 41, cited in the text). The selection of inflammatory response antibodies is based on two factors. Firstly, it's based on previous laboratory studies and, secondly, on the ongoing research project. Alpha-synuclein's role in the brain is interesting because its spatial conformation changes in neurodegenerative diseases, leading to the formation of insoluble accumulation (references 14, 18 and 19, cited in the text), which also happens in the intestine. Both structures have a pro-inflammatory role, and this work shows that exposure to ozone stimulates the accumulation of a specific protein, causing simultaneous inflammatory events. The alpha-synuclein, as described by other authors, plays a role in the inflammatory response, and its folding is irreversible (references 18,19 and 20, cited in the text), which leads to the loss of regulation of the inflammatory response.

2.-In the introduction, the authors present a paragraph about Th17 cells. While it is very likely known to many readers that Th17 cells are linked to IL-17, the authors should first explain why Th17 cells are of interest in this experimental setup.

Answer. I appreciate your assistance. The necessary corrections have already been made in the text.

3.-Most of the expression levels shown in the paper (except IL-17 in substantia nigra) do not follow a linear exposure time path. Assuming increasing tissue O3 levels and very likely also ROS levels, what are the explanations for reduced levels after prolonged exposure, sometimes even to or below control levels (colon IL-17/SRP18). Please explain.

Answer. According to our published research, we observed similar results in mRNA when we measured other proteins after exposing them to ozone. Additionally, other researchers have found that changes in mRNA abundance do not necessarily correspond with changes in protein levels.

Liu et al. have proposed explanations that link the amount of protein available with mRNA's temporal and spatial variations. In our situation, any modifications in the long-term oxidation states and short-term adaptation of the mRNA could play a role. Consequently, the regulation of mRNA for alpha-synuclein might be impacted by the shift in the redox balance within the nucleus and the quantity of misfolded protein in the cytoplasm.

Transcription processes are complex and dynamic, following a series of genomic changes that occur at the nuclear level. These changes include epigenetic modifications due to oxidation, which we have yet to complete our investigation.

4.-Some essential information are missing that need to be added to the text of each according figure: Every depiction of statistical data needs a description in the form of (N (number of independent experiments used for calculation), +/-1,5xIQR (or other whisker definition) and significance indicators (as already provided in current version) (like (N= 6, whisker: 1.5xIQR, p ≤ 0.05 (*). Box blot whiskers are not necessarily defined as 1.5xIQR after Tukey.

Answer. Due to the distribution of the data, we opted for non-parametric tests. Firstly, we used the Kruskal-Wallis test to check if there were any significant differences between all the groups of the different treatments. As we found them to be significant, we further used the Mann-Whitney U test to compare the control group with each of the independent groups of the experiment.

Thank you very much. The missing information in the graphs has been added.

5.- The English language of the manuscript is quite good. However, writing and language should be improved before publication. Some examples are provided in detail below.

line 13: fullstop after ppm

Thank you very much. The change has been made to the text.

line 19:.. increases the presence of alpha-synuclein and induces loss of….

Thank you very much. The change has been made to the text.

line 26: Ozone is not a simple environmental pollutant. Its presence in the troposphere is essential as UV-shield and some O3 is also detected under natural conditions. O3 is however also not the general “main” inducer of ROS. Please rephrase sentence.

Answer. The paragraph has been revised to emphasize the significance of stratospheric ozone.

line 35: Systemic uptake of relevant levels of O3 by “olfactory receptors”directly to the brain needs to be supported by literature.

Answer. The required references were added to the text.

line 42: It is clear that alpha-Syn means alpha-synuclein. Please nevertheless explain every abbreviation at first use.

Thank you very much. The change has been made to the text.

line 45: microglia and dopamine: literature required

Answer. The required references were added to the text.

line 47: induces homooligomerization

Thank you very much. The change has been made to the text.

line 51: that the induced and prolonged inflammation causes neuronal degeneration.

Thank you very much. The change has been made to the text.

line 54: ..the neurodegenerative pathology progresses in the brain.

Thank you very much. The change has been made to the text.

line 77 “emit” is a strange word here

Thank you very much. The change has been made to the text.

line 97,116: Immunhistochemistry revealed an increase of alpha-syn protein after.

Thank you very much. The change has been made to the text.

line 269: In the view that there are many more proteins involved, I would suggest to write “contributing to inflammatory response”

Thank you very much. The change has been made to the text.

line 329: EE.UU ??

Thank you very much. The change has been made to the text.

Round 2

Reviewer 1 Report

Comments and Suggestions for Authors

Dear authors,

-       Statistical processing still concerns me. Mann-Whitney u test is supposed to be used only for comparison of two groups and it does not fit for multiple comparisons. Also, in the notes to Figures you are writing « *= P ≤ 0.05», but you don’t specify between which groups you have found these statistical findings.

-       The conclusion is still too speculative. I highly recommend you not to claim those things without experimental data directly addressing them. Even though it seems logical, you may never be sure that it works this way. Thus, I would recommend you to put all the speculations to the Discussion section. Conclusions should list your key findings and their significance for the field and for future research.

Author Response

Response to reviewer comments and suggestions

Dear Editor,

Once again, we appreciate your corrections. The conclusions were written, and the asterisk was explained in the figure captions following their instructions. We also discussed statistics and provided a reference to support their use.

Thank you very much for helping us improve our work

Dr. Selva Rivas Arancibia

-       Statistical processing still concerns me. Mann-Whitney u test is supposed to be used only for comparison of two groups and it does not fit for multiple comparisons. Also, in the notes to Figures you are writing « *= P ≤ 0.05», but you don’t specify between which groups you have found these statistical findings.

Answer:

Thank you very much; the meaning of the asterisk and the fact that the groups were compared with the control group are explained at the foot of the figure.

Regarding the statistics and the Mann-Whitney U test, I think it wasn't explained clearly. When explaining the meaning of the asterisk, as you requested, it is clear that the samples were independently compared against the control. Also, the foot figure was corrected. It is worth noting that the Mann-Whitney U test is suitable for analyzing small samples in independent groups. To determine which groups differed from the control group, we first compared all groups using the Kruskal-Wallis test. Since the results were significant, we used the Mann-Whitney U test to compare the control group with each one of the groups.

Reference

Graeme D. Ruxton, Guy Beauchamp, Time for some a priori thinking about post hoc testing, Behavioral Ecology, Volume 19, Issue 3, May-June 2008, Pages 690–693, https://doi.org/10.1093/beheco/arn020.

Tex of reference

“The Kruskal–Wallis test is the nonparametric equivalent of 1-way ANOVA. Pairwise multiple comparison procedures based on all possible rank comparisons are available (including Dunnett's T3, Dunnett's C, and Games–Howell tests; Kirk 1995); Toothaker (1993) recommends the last of these and provided a recipe for implementation. Sokal and Rohlf (1995) recommend a simultaneous test procedure, but this requires equal sample sizes in each group. Probably the most commonly used method for pairwise multiple comparisons without making assumptions about normality is the Dunn procedure, as laid out in Zar (1999). Thus, for performing all pairwise comparisons, we would recommend either the Games–Howell procedure (as described by Toothaker 1993) or the Dunn procedure from Zar (1999). For a small set of planned comparisons, we would recommend Mann–Whitney U-tests with control of EER using one of the methods discussed above for parametric comparisons”. Graeme D. Ruxton et al 2008

-       The conclusion is still too speculative. I highly recommend you not to claim those things without experimental data directly addressing them. Even though it seems logical, you may never be sure that it works this way. Thus, I would recommend you to put all the speculations to the Discussion section. Conclusions should list your key findings and their significance for the field and for future research.

Answer:

Thank you so much for the correction. You are right about the conclusions. Therefore, the text was moved to the end of the discussion, and we wrote other conclusions based on our results.
